# PrecogUI: Proactive GUI agents via Pre-cognitive Simulation and Experience Retrieval

## Abstract

Existing reactive Graphical User Interface (GUI) agents often fail in long-horizon, dynamic scenarios, where unexpected disturbances trigger attention-diverting and cascading failures. To address this, we propose **PrecogUI**, a pre-cognitive architecture that shifts the paradigm from reactive execution to proactive decision-making. Specifically, we first design a Proactive Experience Pool (PEP) which caches frequently occurring anomaly and success patterns as "state-action-result" tuples in a graph structure, forming a composable prior memory. Furthermore, we introduce a Proactive Simulation Executor (PSE) that learns to forecast the next symbolic UI layout given a candidate action, enabling early anomaly avoidance and estimating action success probabilities. Finally, a Pre-cognitive Execution Controller (PEC) fuses these priors and predictions, prioritizes handling of foreseen anomalies, and ensures execution robustness through a closed-loop error correction mechanism. For robust evaluation, we develop AutoTraj, an automatic data-generation engine, to construct InterfereBench, a benchmark for long-horizon tasks with strong disturbances. Experiments demonstrate that PrecogUI surpasses existing state-of-the-art methods on InterfereBench while maintaining competitive performance on public benchmarks. The code and models will be publicly available.

## 1 Introduction

Graphical User Interface (GUI) Agents (Cheng et al., 2024; Lin et al., 2025; Gou et al., 2025a; Hong et al., 2024) are built on Multi-modal Large Language Models (MLLMs) to comprehend user queries, interpret context, and perform actions like clicks and swipes for accomplishing GUI tasks. The advancement of MLLMs (Li et al., 2023; Alayrac et al., 2022; Dai et al., 2023a) has notably enhanced agents' interface perception and decision-making precision. Nevertheless, disruptive anomalies such as pop-ups and black screens in dynamic settings persist as a significant hurdle, diverting attention and leading to cascading errors with prolonged consequences.

Prior research (Hong et al., 2024; Huang et al., 2025; Chen et al., 2025) has significantly advanced the perception-action loop. However, the prevailing approach remains reactive, relying on current observations for decision-making. While effective in short-horizon, disturbance-free settings (Rawles et al., 2025; Deng et al., 2023), these reactive methods may struggle in long-horizon tasks and dynamic environments. Recent efforts have attempted to address this challenge through online exploration (Sun et al., 2025; Fan et al., 2025) and improved visual-layout alignment robustness (Wen et al., 2024b; Kong et al., 2025). Nevertheless, the reactive nature still leaves agents vulnerable to distractions from non-goal cues such as pop-ups and loading delays.

**Key Observations.** To investigate the robustness, we evaluate representative reactive agents (Liu et al., 2025a; Qin et al., 2025; Zhang et al., 2025b) on AndroidControl (Li et al., 2024) under injected disturbances at both the overlay level (e.g., pop-ups, notifications) and environment level (e.g., black screens, freezing). Performance is assessed by success rate (SR), stratified by disturbance type and task horizon. Specifically, as shown in Figure 1(b), the overlay-level disturbances induce the most significant degradation, reducing SR by over 20% on average, compared to a milder 10% drop under environment-level perturbations. Besides, the performance degradation scales monotonically

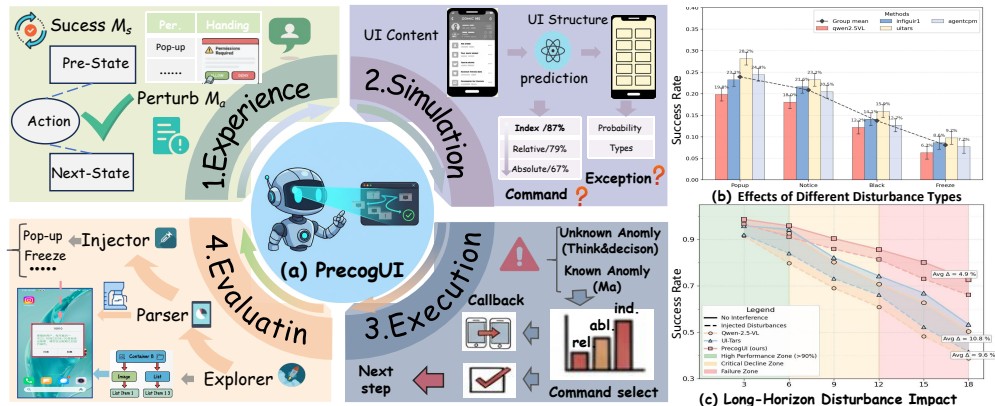

Figure 1: (a) PrecogUI: a pre-cognitive architecture integrating prior experience with look-ahead simulation; AutoTraj provides perturbed trajectories. (b) Disturbance-type sensitivity on InterfereBench. (c) Disturbance effects amplify with task horizon.

with horizon length, as shown in Figure 1(c). On short-horizon tasks ($< 5$ steps), all models maintain high robustness ($SR \geq 91\%$). However, for medium-length tasks (6–15 steps), reactive agents exhibit relative SR drops of about 10–14%. In long-horizon tasks (>15 steps), reactive agents SR declines to about $40\%$.

Given the above results, we can observe that the reactive agents are prone to distraction by non-goal-related stimuli, leading to potential cascading failures as errors accumulate over trajectories. This observation prompts a crucial question: *how can we empower agents with pre-cognitive planning and explicit exception handling to ensure the robustness in long-horizon, dynamic environments?*

**Our Solution.** In this study, we propose PrecogUI, a framework that integrates experience retrieval with online look-ahead simulation to improve the robustness of GUI agents in long-horizon and disturbance-prone settings. The conceptual architecture is illustrated in Figure 1(a).

Specifically, PrecogUI introduces the Proactive Experience Pool (PEP), a graph-structured repository that stores and retrieves recurring interaction patterns from both successful and anomalous executions, enabling knowledge reuse via historical pattern matching. Then, the Proactive Simulation Executor (PSE) employs a conditional diffusion model (Rombach et al., 2022) to simulate the symbolic UI layout resulting from candidate actions, providing lookahead forecasts for early anomaly detection and action success estimation. Finally, these are integrated by the Pre-cognitive Execution Controller (PEC), which prioritizes anomaly handling, selects high-utility actions, and ensures robustness through state monitoring and hierarchical rollback/retry.

To the best of our knowledge, no existing benchmark evaluates long-horizon robustness under realistic perturbations. We thus introduce InterfereBench, a new benchmark consisting of 1,160 long-horizon trajectories (27k screenshots) across 34 diverse applications. These trajectories incorporate prolonged task horizons and intense, dynamic interferences, and AutoTraj, an automated engine for generating diverse, perturbation-rich interaction trajectories at scale. Experiments on InterfereBench and generic benchmarks like AndroidControl and GUI-Odyssey show that PrecogUI significantly outperforms the baseline by 22.1% in success rate under strong perturbations, improving robustness without sacrificing overall performance. To summarize, our contributions are as follows:

- We propose PrecogUI, a unified framework that combines offline experience reuse, proactive layout prediction, and exception-aware execution recovery to enhance robustness in long-horizon GUI interactions.

- We present InterfereBench, a new benchmark designed to evaluate robustness under strong, sustained perturbations in long-horizon tasks, along with AutoTraj, an automated pipeline for scalable, realistic trajectory generation.

- Extensive experiments on InterfereBench and the public benchmarks (AndroidControl (Li et al., 2024) and GUI-Odyssey (Lu et al., 2024)) demonstrating that PrecogUI effectively improves long-horizon reliability and anomaly resilience, while still maintaining the general capabilities.

Figure 2: Data Construction. Stage 1 discovers clickable elements via view hierachy and vision, executes basic actions, and logs replayable UI trajectories. Stage 2 prunes redundant steps, ranks trajectories with an MLLM, and outputs structured annotations. Stage 3 injects realistic disturbances to create cleanperturbed pairs for robustness evaluation.

## 2 METHOD

### 2.1 OVERVIEW

Toward robust long-horizon execution under perturbations, we propose PrecogUI, which closes the loop between experience, foresight, and feedback via four modules: (i) PEP forms a graph-structured memory of anomaly/success patterns via layout hashing and nearest-neighbor retrieval; (ii) PSE predicts the next symbolic UI layout, estimates anomaly risk, and ranks candidate actions (index/relative/absolute); (iii) PEC fuses PEP and PSE with online monitoring and rollback/retry to deliver robust, closed-loop control; and (iv) AutoTraj builds InterfereBench, a long-horizon benchmark with controlled perturbations. The discussion of related work is in the Appendix 11 due to the page limit.

### 2.2 DATA CONSTRUCTION

The capabilities of GUI agents are fundamentally constrained by data scale, diversity, and quality. To address this, we present AutoTraj, an automated pipeline that generates high-quality GUI interaction trajectories with explicit disturbance awareness. AutoTraj comprises three core components:

**Autonomous Explorer.** The Explorer efficiently discovers diverse, high-value interaction trajectories using a hybrid perception strategy: it prefers UI view hierarchy to find actionable elements, when structured signals are missing or incomplete, it falls back to a vision pipeline that combines object detection and optical character recognition (OCR), producing a unified candidate set of controls.

Exploration is driven by a pre-trained agent (Ye et al., 2025) that sequentially tries atomic actions (click, scroll) and logs pre- and post-screenshots, as well as action metadata, to produce replayable trajectories. To guide informative exploration, we define the exploration value at state $s_t$ as:

$$V(s_t) = \alpha \cdot \frac{\left| E_t \setminus \left( \bigcup_{i<t} E_i \right) \right|}{|E_t| + \varepsilon} + (1-\alpha) \cdot \frac{1}{\sqrt{n(s_t) + 1}}, \tag{1}$$

where $E_t$ denotes the control set at $s_t$, $\bigcup_{i<t} E_i$ is the union of controls seen so far, and $n(s_t)$ counts visits to $s_t$. The first term promotes the discovery of unseen controls/layouts, while the second enforces novelty to favour coverage and rarely visited states. $\alpha \in [0, 1]$ balances layout discovery and rare-state exploration; hyperparameter analysis appears in Appendix 11.

**Trajectory Parser.** To ensure semantic and structural quality, the raw trajectories undergo a two-stage filtering and parsing process. Stage-1 removes traces with excessive length and redundancy, we formalise $\rho_{\text{loop}}$ and $\rho_{\text{noop}}$ as follows:

$$\rho_{\text{loop}} = \frac{1}{T} \sum_{t=1}^{T} \mathbf{1}[a_t = \text{self-loop}], \qquad \rho_{\text{noop}} = \frac{1}{T} \sum_{t=1}^{T} \mathbf{1}[\mathcal{D}_{\text{layout}}(L_t, L_{t+1}) < \tau_c]. \tag{2}$$

where, $T$ is the number of steps, $a_t$ is the action at step $t$, $L_t$ is the UI layout at step $t$, $\mathbf{1}[\cdot]$ denotes the indicator function, $\mathcal{D}_{\text{layout}}$ is the layout-difference measure, and $\tau_c$ is the no-op threshold. If either

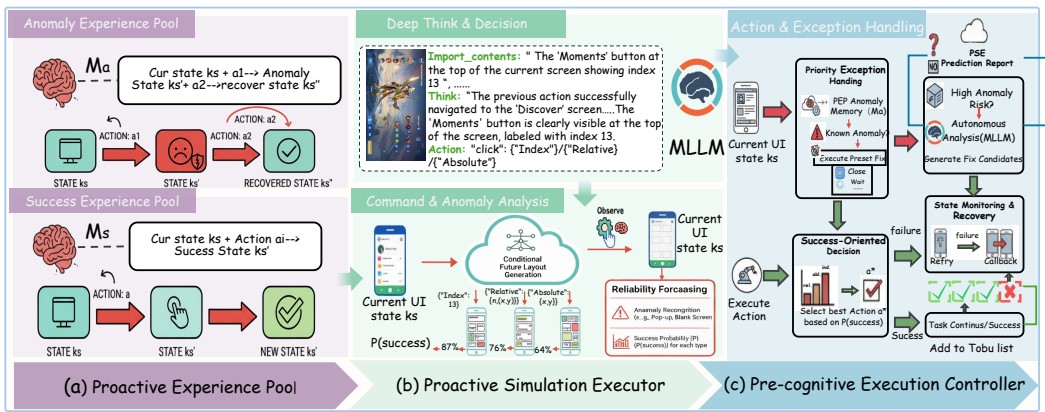

Figure 3: Illustration of PrecogUI. (a) PEP builds a graph memory of anomaly/success patterns; (b) PSE forecasts the next symbolic UI layout and estimates anomaly risk for candidate actions; (c) PEC fuses priors and predictions to prioritize exception handling, select the highest-utility action, and enforce closed-loop monitoring with rollback/retry.

ratio exceeds its preset threshold, the trajectory is pruned as structurally low-quality. Further details are provided in Appendix 11

In Stage-2, we further filter trajectories using a high-capacity MLLM (Comanici et al., 2025b) that evaluates topic consistency, causal soundness, and task complexity, selecting the top-$K$ rated trajectories for detailed labelling. For each trajectory, the parser yields a high-level task goal and stepwise descriptions, and exports structured JSON containing goals, step descriptions, action types (normalised coords), UI boxes, screen deltas, and execution outcomes for training and evaluation.

**Perturbation Injector.** To study robustness, we develop a Perturbation Injector that creates paired samples for comparative evaluation. For each clean trajectory, we randomly inject real-world perturbations covering: (1) overlay interference (simulating system notifications, pop-up dialogues, etc.); (2) environmental perturbations (black or repeated frames for loading/lag, layout changes). This process yields paired samples for each trajectory: a clean baseline and perturbed variants, enabling comparative evaluation in both normal and perturbed modes.

Following this pipeline, AutoTraj produces the InterfereBench benchmark covering over 34 application scenarios, with 1,160 long-horizon trajectories (14–37 steps each) and around 27k annotated screenshots. Each trajectory includes one clean baseline and two perturbed variants of different types, providing a solid foundation for a comprehensive evaluation of GUI agents' robustness in long-horizon, dynamic environments.

To faithfully capture and replay complex interactions beyond single-tap actions, we additionally employ **PolyTouch** (Sec. 11), a multi-gesture and macro execution layer that synthesizes deterministic multi-pointer gestures (e.g., three/four-finger chords, pinch/zoom/rotation) and declarative macros with explicit timing, guards, retries, and rollback.

## 2.3 PROACTIVE EXPERIENCE POOL

We observe that failure-inducing anomaly patterns (e.g., permission pop-ups, network delays) and success-inducing patterns (e.g., app navigation) repeat widely across tasks and applications. Therefore, we propose the Proactive Experience Pool (PEP), which converts costly trial-and-error into efficient experience retrieval. By caching and indexing critical stateactionoutcome patterns, the executor can leverage priors rather than plan in isolation. PEP maintains two parallel memories:

(1) *Anomaly Memory ($M_a$):* $M_a$ records two classes of failures: (i) stateaction mappings $(k_s, k_a) \mapsto \ell_{\text{anom}}$ when an action in a state yields a specific anomaly; (ii) $k_s \mapsto \ell_{\text{anom}}$ for states that inherently denote failure (e.g., network outage).

(2) *Success Memory ($M_s$):* $M_s$ maintains high-confidence transitions as successful transitions $(k_s, k_a) \mapsto k_{s'}$, denoting that action $a$ in state $s$ reliably reaches a successful successor state $s'$.

**State Representation and Retrieval Mechanism.** To enable robust and efficient retrieval, we represent each UI state $s$ by a canonical layout hash $k_s = h_s(L_t)$ (hash-based). This key is computed from a deterministically sorted list of interactive elements, where each element is abstracted as *(type, bbox)*. Similarly, actions $a$ are canonicalized to a key $k_a$ based on their type and the target element's normalized coordinates.

Crucially, PEP is a dynamic, online-updated memory. Upon encountering new anomalies or discovering successful cases via exploration, the agent extracts the associated state-action-result tuple in real time and asynchronously appends it to the memory pool.

## 2.4 PROACTIVE SIMULATION EXECUTOR

The mainstream GUI agents are fundamentally reactive, lacking foresight into post-action effects. With abrupt transitions in dynamic UIs, reactive policies without anticipation of anomalies tend to fall into irrecoverable failures. Accordingly, we propose the Proactive Simulation Executor (PSE), which forecasts the next symbolic UI layout before acting, and evaluates anomaly risk and candidate action's success probability, shifting the paradigm from observe–act to observe–predict–act.

**Conditional Future Layout Generation.** To capture the inherent uncertainty of UI state transitions, we model the next-step layout distribution $P(\hat{L}_{t+1}|L_t, a_t, g)$ using a conditional latent diffusion model (Rombach et al., 2022), which is fine-tuned on a comprehensive dataset comprising InterfereBench and several public GUI datasets (Li et al., 2024; Lu et al., 2024). For computational efficiency, we abandon pixel-space generation and predict a symbolic layout $L_{t+1}$, represented as an ordered set of (type, bbox). It is conditioned on a joint embedding $\mathbf{C}$ of the current layout $L_t$, the candidate action $a_t$ and the task goal $g$.

$$\mathbf{l}_{i-1} = \frac{1}{\sqrt{\alpha_i}} \left( \mathbf{l}_i - \frac{1-\alpha_i}{\sqrt{1-\bar{\alpha}_i}} \epsilon_\theta(\mathbf{l}_i, i, \mathbf{C}) \right) + \sigma_i \epsilon \tag{3}$$

where $\alpha_i$, $\bar{\alpha}_i$, and $\sigma_i$ are the pre-specified noise-schedule parameters. Finally, the denoised latent $\hat{\mathbf{l}}_{t+1}$ is decoded into a discrete symbolic layout, yielding the predicted future layout $\hat{L}_{t+1}$.

**Reliability Forecasting.** Subsequently, we apply a set of efficient rules to the predicted layout $\hat{L}_{t+1}$ for anomaly recognition. For example, if a bounding box significantly occludes multiple interactive controls in $L_t$, it is flagged as a pop-up Anomaly; if interactive elements are nearly absent, a Blank-Screen Anomaly; if $\hat{L}_{t+1}$ remains largely unchanged from $L_t$, the action is likely ineffective or causes freezing.

Our premise is that success correlates with salient layout shifts. To quantify this, for each candidate action $a_t$, we introduce the Layout Dissimilarity Score, formalized as:

$$\mathcal{D}_{\text{layout}}(L_t, \hat{L}_{t+1}) = 1 - \frac{2|\mathcal{M}|}{|L_t| + |\hat{L}_{t+1}|} \tag{4}$$

where $\mathcal{M}$ is the set of matched pairs between the $L_t$ and $\hat{L}_{t+1}$. This score is naturally normalized: it is 0 for identical layouts and approaches 1 for completely dissimilar ones. We then directly use this dissimilarity score as the success probability, as it reflects the magnitude of the predicted UI transformation:

$$P(\text{success} \mid a_t, \text{type}) = \mathcal{D}_{\text{layout}}(L_t, \hat{L}_{t+1}) \tag{5}$$

Finally, PSE returns a comprehensive reliability report for each candidate action, including anomaly probabilities and type-specific success probabilities, to the downstream decision-making module.

## 2.5 PRE-COGNITIVE EXECUTION CONTROLLER

While PSE offers look-ahead predictions, robustness remains uncertain in the absence of a decision framework that converts them into concrete actions. We design the Pre-cognitive Execution Controller (PEC), a closed-loop controller that fuses PEP priors, PSE predictions, and execution feedback, converting open-ended trial-and-error into guided, self-correcting policy control.

---

**Algorithm 1** Pre-cognitive Execution Controller

---

**Require:** Goal $g$, current state $s_t$, anomaly memory $M_a$
**Ensure:** A single successful action-type pair $(a, \text{type})$ or a fallback diagnosis action
    *// – Stage 1: Pre-cognitive Anomaly Checks –*
1:   $k_s \leftarrow \text{Hash}(\text{Layout}(s_t))$
2:   **if** $M_a.\text{Query}(k_s)$ returns a remedy $a_{\text{handle}}$ **then**
3:      **return** $a_{\text{handle}}$
4:   **if** $\text{PSE.ForecastRisk}(s_t) > \tau_{\text{risk}}$ **then**
5:      **return** $\pi_{\text{LLM}}(s_t, \mathcal{T}_{\text{anomaly}})$                  ▷ Autonomous diagnosis for novel anomaly
    *// – Stage 2: Iterative Execution and Recovery Loop –*
6:   $\mathcal{A} \leftarrow \text{MLLM.GenerateCandidates}(s_t, g)$
7:   $\mathcal{F}_t \leftarrow \emptyset$                                           ▷ Initialize temporary taboo list
8:   **while** $\mathcal{A}$ is not empty **do**
9:      $(a^*, \text{type}^*) \leftarrow \arg\max P(\text{success} \mid a_i, \text{type}_j)$
10:     **execute** $(a^*, \text{type}^*)$; **observe** new state $s_{t+1}$
11:     **if** $\text{MLLM.VerifySuccess}(s_t, (a^*, \text{type}^*), s_{t+1})$ **then**
12:       **return** $(a^*, \text{type}^*)$                         ▷ **Success**: terminate step
13:      $\mathcal{F}_t \leftarrow \mathcal{F}_t \cup \{(a^*, \text{type}^*)\}$
14:     **if** $\mathcal{D}_{\text{layout}}(\text{Layout}(s_t), \text{Layout}(s_{t+1})) < \tau_c$ **then**          ▷ Stagnation
15:       **continue**
16:     **else**                                    ▷ Unexpected Transition
17:       ROLLBACK
18:     **if** all types for $a^*$ are in $\mathcal{F}_t$ **then**
19:       $\mathcal{A} \leftarrow \mathcal{A} \setminus \{a^*\}$
20: **return** $\pi_{\text{LLM}}(s_t, \mathcal{T}_{\text{anomaly}})$            ▷ Final diagnosis if all candidates fail

---

**Deep Think & Decision.** The cycle begins by generating a set of semantically grounded candidate actions $\mathcal{A}$. We steer a base MLLM's reasoning by prompting it to populate a structured JSON schema. This schema mandates a chain of thought that includes: (i)Historical Validation, verifying the outcome of the previous step; (ii)Content Grounding, ensuring that critical UI elements for the current instruction are present; (iii)Think, a step for rationale articulation and failure attribution analysis; and finally (iv)Action, which outputs a ranked set of candidate actions, each with index, relative, and absolute coordinate formats, further details are provided in Appendix 11.

**Pre-cognitive Decision and Execution.** Prior to execution, PEC performs a pre-execution anomaly check on $s_t$. It first queries the anomaly memory $M_a$ with the layout hash $k_s$, if it matches a known anomaly, the controller triggers the preset remedy $a_{\text{handle}}$; If the lookup fails and PSE indicates elevated risk, PEC activates autonomous diagnosis and uses a foundation model with a structured anomaly prompt (details in Appendix 11) to synthesize an immediate handling action $a_{\text{handle}}$.

When the state is judged as normal, PEC proceeds with a success-oriented decision. It takes the candidate actions $\mathcal{A}$ from the previous stage and requests reliability reports from PSE. Finally, it selects and executes the action-type pair $(a^*, \text{type}^*)$ according to:

$$(a^*, \text{type}^*) = \underset{a_i \in \mathcal{A}}{\text{argmax}} \left( P(\text{success} \mid a_i, \text{type}_j) \right) \tag{6}$$

The selection is constrained to actions that are not flagged as high-risk by PSE.

**State Monitoring & Adaptive Recovery** After executing the action $(a^*, \text{type}^*)$, PEC captures the new state $s_{t+1}$ and verifies the outcome via both the layout change $\mathcal{D}_{\text{layout}}(L_t, L_{t+1})$ and a semantic validation from its MLLM, as part of the subsequent step's Historical Validation. If the execution is judged a failure, PEC triggers recovery according to the failure type:

- *Stagnation*: For minimal layout change ($\mathcal{D}_{\text{layout}}(L_t, L_{t+1}) < \tau_c$), PEC treats the instruction as invalid and retries the action using PSEs next-best instruction type.

Table 1: Performance on InterfereBench under two instruction settings (Low/High), $TM_n$ and $SR_n$ denote type-matching and success rate on the normal (non-perturbed) subset; and $TM_a$ and $SR_a$ denote the corresponding metrics on the perturbed subset.

| Method | InterfereBench-Low | | | | InterfereBench-High | | | |
|---|---|---|---|---|---|---|---|---|
| | $TM_n$ | $SR_n$ | $TM_a$ | $SR_a$ | $TM_n$ | $SR_n$ | $TM_a$ | $SR_a$ |
| GPT-4o | 76.7 | 22.1 | 69.0 | 9.3 | 73.4 | 3.9 | 65.0 | 1.2 |
| Gemini-2.5 | 87.1 | 28.4 | 80.0 | 12.5 | 83.8 | 17.7 | 76.0 | 7.5 |
| Qwen-2.5-VL | 90.6 | 33.7 | 82.5 | 18.9 | 71.7 | 36.5 | 64.0 | 18.0 |
| OmniParser | 84.1 | 70.9 | 76.0 | 41.5 (↓29.4) | 70.6 | 27.3 | 63.0 | 12.8 (↓14.5) |
| InfiGUI-R1 | 88.9 | 73.6 | 81.5 | 45.8 (↓27.8) | 77.4 | 37.3 | 68.0 | 19.5 (↓17.8) |
| OS-Atlas | 88.4 | 72.1 | 80.8 | 43.3 (↓28.8) | 72.8 | 30.4 | 63.5 | 14.7 (↓15.7) |
| AgentCPM | 90.0 | 75.7 | 82.1 | 46.0 (↓29.7) | 78.9 | 35.7 | 66.8 | 18.4 (↓17.3) |
| UI-TARS-1.5 | 90.8 | 74.5 | 82.6 | 45.0 (↓29.5) | 76.9 | 36.6 | 66.0 | 19.2 (↓17.4) |
| **PrecogUI** | **92.9** | **79.2** | **90.0** | **68.4** (↓10.8) | **80.0** | **52.7** | **74.5** | **41.6** (↓11.1) |

Table 2: Grounding Performance Comparison on the ScreenSpot Benchmark.

| Method | Mobile | | Desktop | | Web | | Avg |
|---|---|---|---|---|---|---|---|
| | *Text* | *Icon* | *Text* | *Icon* | *Text* | *Icon* | |
| GPT-4o | 30.5 | 23.2 | 20.6 | 19.4 | 11.1 | 7.8 | 18.8 |
| Gemini2.0 | – | – | – | – | – | – | 84.0 |
| Qwen-2.5-VL | – | – | – | – | – | – | 84.7 |
| CogAgent | 67.0 | 24.0 | 74.2 | 20.0 | 70.4 | 28.6 | 47.4 |
| SeeClick | 78.0 | 52.0 | 72.5 | 30.0 | 55.7 | 32.5 | 53.4 |
| ShowUI | 92.3 | 75.5 | 76.3 | 61.1 | 81.7 | 63.6 | 75.1 |
| OmniParser | 93.9 | 57.0 | 91.3 | 63.6 | 81.3 | 51.0 | 75.1 |
| UI-Tars-1.5 | 93.0 | 75.5 | 90.7 | 68.6 | 84.3 | 74.8 | 82.3 |
| InfiGUI-R1 | **97.1** | 81.2 | 94.3 | 77.1 | 91.7 | 77.6 | 87.5 |
| **PrecogUI** | 96.5 | **87.8** | **97.5** | **82.2** | **94.6** | **91.7** | **91.2** |

- *Unexpected Transition*: If the layout changes significantly ($\mathcal{D}_{\text{layout}}(L_t, L_{t+1}) \geq \tau_c$) but the MLLM's semantic validation deems the new state an incorrect outcome, PEC performs a rollback and adds the failed pair to a temporary taboo list $\mathcal{F}_t$ to block immediate reuse.

Overall, PEC not only executes optimally in normal settings but also adapts to failures, boosting robustness and recovery in uncertain environments.

## 3 EXPERIMENTS

We implement PrecogUI on a smartphone UI-automation stack, using Gemini-2.5-Pro as the reasoning backend (Comanici et al., 2025a). The system is zero-training end-to-end: all modules are non-learned except the PSEs next-step layout predictor, which is lightly fine-tuned on InterfereBench and public datasets to model action-conditioned UI transitions. More implementation details are in Appendix 11.

To evaluate model performance, we use two types of datasets: (1) Self-constructed InterfereBench, designed to test agent robustness in long-horizon and disturbance tasks. It includes two settings: (i) normal, a clean environment; (ii) perturbation, with dynamic perturbations like overlays and layout changes. (2) Public benchmarks, split into (a) ScreenSpot (Cheng et al., 2024), which measures UI element grounding (text/icon) to assess localization capability, and (b) navigation-centric suites AndroidControl (Li et al., 2024) and GUI-Odyssey (Lu et al., 2024), which evaluate end-to-end task completion and generalization under Low/High instruction settings. Evaluation follows standard GUI agent metrics: success rate (SR) and type matching (TM).

Table 3: Performance comparison on AndroidControl and GUI-Odyssey benchmarks

| Method | AndroidControl-Low | | AndroidControl-High | | GUI-Odyssey | | Avg |
|---|---|---|---|---|---|---|---|
| | *TM* | *SR* | *TM* | *SR* | *TM* | *SR* | |
| GPT-4o | 74.3 | 19.4 | 63.1 | 21.2 | 37.5 | 5.4 | 36.8 |
| Qwen-2.5-VL | 94.1 | 85.0 | 75.1 | 62.9 | 59.5 | 46.3 | 70.5 |
| UI-Tars-1.5 | **98.0** | **91.3** | 83.7 | 72.5 | **94.6** | 87.0 | 87.4 |
| SeeClick | 93.0 | 75.0 | 82.9 | 59.1 | 71.0 | 53.9 | 72.5 |
| Aria-UI | – | 67.3 | – | 10.2 | – | 36.5 | – |
| OS-Atlas | 91.9 | 80.6 | 84.7 | 67.5 | 83.5 | 56.4 | 77.4 |
| InfiGUI-R1 | 96.0 | 92.1 | 82.7 | 74.4 | – | – | – |
| AgentCPM-GUI | 94.4 | 90.2 | 77.7 | 69.2 | 90.9 | 75.0 | 82.9 |
| **PrecogUI** | 94.9 | 88.7 | **86.8** | **76.4** | 91.3 | **89.1** | **87.7** |

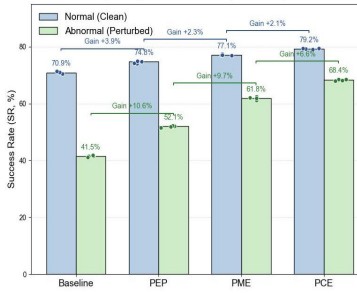

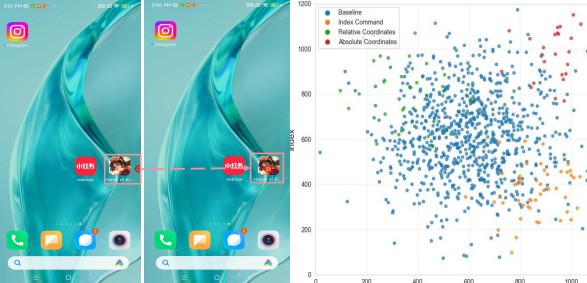

(a) PrecogUI Component Ablation  (b) Failure correction for actions command

Figure 4: Ablation and reliability analysis of PrecogUI.

## 3.1 MAIN RESULTS

**Perturbation Handling.** We evaluate PrecogUI on InterfereBench to assess long-horizon and dynamic-UI robustness. We compare against: 1) General base models, GPT-4o (OpenAI et al., 2024), Gemini-2.5 (Comanici et al., 2025b), and Qwen-2.5-VL (Bai et al., 2025); 2) Specialized GUI agents: OmniParser (Wan et al., 2024), InfiGUI-R1 (Liu et al., 2025b), AgentCPM, OS-Atlas (Wu et al., 2024), and UI-TARS. As shown in Table 1, on the normal subset, PrecogUI reaches an SR of 79.2% (Low) and 52.7% (High), outperforming the best reactive GUI agent by 3.5% and 15.4%, respectively. Crucially, on the perturbation subset, PrecogUI shows much less degradation by only 10.8% / 11.1% (Low/High), and surpasses previous SOTA by 23.4% (Low) and 22.4% (High).

**Grounding Capability.** Table 2 reports ScreenSpot grounding accuracy across mobile/desktop/web. PrecogUI consistently outperforms prior methods across device subsets (mobile, desktop, web), achieving 91.2% average grounding accuracy, surpassing all open-source models and outperforming the previous SOTA (InfiGUI-R1) GUI framework by 3.7%.

**Navigation Capability.** To validate generalization, PrecogUI is evaluated on mainstream public benchmarks, including AndroidControl and GUI-Odyssey. As shown in Table 3, on AndroidControl, PrecogUI achieves success rates of 88.7% (Low) and 76.4% (High), and reaches 89.1% on GUI-Odyssey. PrecogUI surpasses a base model by 8.2% and exceeds an instruction-tuned, GUI-specialised agent by an average of 4.5%. The findings support that PrecogUI is a general, efficient decision paradigm rather than a solution limited to particular disturbances.

## 3.2 ABLATION STUDY

**Effectiveness of the Experience Pool (PEP).** To assess the contributions of PrecogUI components, we conduct ablations on the InterfereBench strong-perturbation subset. As shown in Figure 4(a) A purely reactive baseline achieves 41.5% SR that relies only on the base model. With PEP integrated, SR improves by 14.6%, showing the benefit of layout-based hashing to reuse success/anomaly pat-

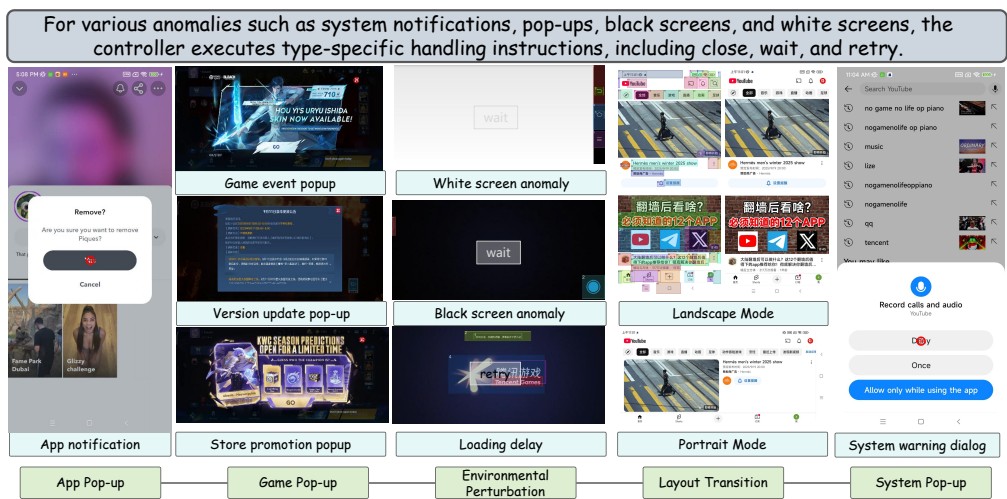

Figure 5: Case Visualization of PrecogUI in Various Anomalous Scenarios

terns for trap avoidance. When used without PEP, PSE increases SR by 11.3%, demonstrating the utility of forecasting the next layout to preempt unseen disturbances. With all components enabled, PrecogUI reaches 68.4% SR. On the clean long-horizon subset, components also provide gains, with SR rising from 70.9% to 79.2%. These results demonstrate the robustness of PrecogUIs components on long-horizon, strongly perturbed tasks.

**Analysis of Proactive Reliability Forecasting.** PSE is designed to forecast the next-frame layout transition for index, relative, and absolute instructions individually, providing an estimate of their expected return. We compare two strategies: (i) a fixed policy that always uses a single instruction type (e.g., index only); (ii) a dynamic policy that executes the instruction with the highest expected return predicted by PSE. As shown in Figure 4(b), the fixed index policy is fragile in dynamic UIs, where pop-ups or layout shifts reduce its SR to only 74.8%. Under the normal mode, the dynamic strategy recovered 40 failed trajectories and improved step-wise SR to 79.2%. This gain confirms that in dynamic GUIs, no single instruction type remains optimal across contexts.

### 3.3 CASE VISUALIZATION

As illustrated in Figure 5, PrecogUI demonstrates strong anomaly-handling capabilities across cross-application, strong-perturbation, and dynamic GUI tasks. Whether facing common application or in-game pop-ups, extreme disturbances such as black or white screens, or drastic layout changes with high-priority system-level pop-ups, PrecogUI consistently shows distinct advantages. In addition to dealing with expected pop-ups, PrecogUI relies on predictive modeling to foresee layout shifts and latent abnormal states. For example, under loading delays, the model anticipates the disruption and chooses to wait, avoiding erroneous interactions with empty or unresponsive screens. This foresight-driven anomaly-avoidance mechanism ensures robust execution in complex task flows, a capability absent in reactive agents.

## 4 CONCLUDING REMARKS

In this work, we address the core challenge that reactive policies in long-horizon, disturbance-rich GUI tasks are easily diverted by anomalies and lead to cascading failures by introducing PrecogUI, a pre-cognitive framework that shifts decision-making from reactive control to anticipatory reasoning. PrecogUI follows an experienceforesightfeedback loop: PEP encodes anomaly and success patterns into retrievable priors via layout hashing; PSE forecasts the next symbolic layout and evaluates risks and success likelihood; PEC fuses these priors and predictions with runtime monitoring and rollback/retry, delivering anomaly-first yet success-driven closed-loop control. Finally, we validate PrecogUI on both our constructed InterfereBench and public benchmarks, demonstrating significant performance gains and robust execution in complex, dynamic environments.

ETHICS CHECKLIST

**1. Code of Ethics Acknowledgement**

1.1. All authors have read and will adhere to the ICLR Code of Ethics; acknowledgement was made during submission (yes/no) yes

1.2. This paper includes an Ethics Statement at the end of the main text, before references (if applicable) (yes/no) yes

**2. Human Subjects and IRB/Consent**

2.1. Research involves human subjects or user studies (yes/no) NA

 If yes, address the following:

2.2. IRB/ethics board approval (or equivalent) is obtained and documented (yes/NA) NA

2.3. Informed consent procedures are described; compensation and inclusion of minors are disclosed (yes/NA) NA

**3. Data, Privacy, and Security**

3.1. All datasets used are cited with licenses and access conditions; non-public data are described with justification (yes/partial/no) yes

3.2. Personally identifiable information (PII) was removed, anonymized, or processed under compliant safeguards (yes/NA) yes

3.3. Data collection respects terms of service and legal/compliance requirements (e.g., copyright, web scraping policies) (yes/partial/no) yes

3.4. Security-sensitive artifacts or vulnerabilities are responsibly handled (e.g., redactions, coordinated disclosure) (yes/NA) NA

**4. Bias, Fairness, and Potential Harm**

4.1. Known risks of harmful or dual-use applications are discussed with mitigation strategies (yes/partial/no) yes

4.2. Bias/fairness concerns (subgroup performance, demographic or domain skews) are analyzed or acknowledged (yes/partial/no) partial

4.3. Limitations, open risks, and appropriate use/disallowed use are stated (yes/no) yes

**5. Conflicts of Interest and Sponsorship**

5.1. All funding sources, compute donations, and in-kind support are disclosed (yes/no) yes

5.2. Potential conflicts of interest (employment, consulting, equity) are disclosed (yes/NA) NA

**6. Research Integrity**

6.1. All results are reported faithfully; negative findings or failure cases are included when relevant (yes/partial/no) yes

6.2. Figures/tables are accurately labeled; data provenance and documentation are maintained (yes/-partial/no) yes

*Note: The Ethics Statement is optional but recommended; it does not count toward the page limit and should not exceed one page.*

## REPRODUCIBILITY CHECKLIST

### 7. Overall Documentation

7.1. High-level method overview and/or pseudocode provided (yes/partial/no) yes

7.2. Clear separation of claims vs. evidence; notation and assumptions are stated (yes/partial/no) yes

7.3. Pointers to background/pedagogical resources for replication (yes/no) yes

### 8. Code, Artifacts, and Environment

8.1. Anonymous, downloadable code provided as supplementary material or link (yes/partial/no) yes

8.2. Exact commit/version, dependency list (e.g., `environment.yml`/`requirements.txt`), and OS details (yes/partial/no) yes

8.3. Hardware details (GPU/CPU models, RAM), framework/library versions, and runtime estimates (yes/partial/no) yes

8.4. Randomness handling documented (seeds, nondeterministic ops, determinism limits) (yes/partial/no/NA) yes

### 9. Data and Preprocessing

9.1. All datasets cited with URLs/licensing; custom splits or filtering rules documented (yes/partial/no) yes

### 10. Training and Hyperparameters

10.1. Search spaces and selection criteria reported; final hyperparameters listed per model (yes/partial/no) yes

10.2. Training schedules, batch sizes, losses, and early-stopping criteria documented (yes/partial/no) yes

### 11. Evaluation and Reporting

11.1. Metrics are formally defined and motivated; evaluation scripts included (yes/partial/no) yes

11.2. Number of runs, variance (e.g., std/CI), and significance tests reported where appropriate (yes/-partial/no) partial

11.3. Ablations/diagnostics provided to support claims and clarify failure modes (yes/partial/no) yes

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

## APPENDIX

This is the supplementary file for our submission titled *PrecogUI: Proactive GUI agents via Pre-cognitive Simulation and Experience Retrieval.* This material supplements the main paper with the following content:

## THE USE OF LARGE LANGUAGE MODELS

In this work, large language models (LLMs) are used exclusively for polishing the writing and checking grammar. They are not involved in research ideation, experimental design, data analysis, or the formulation of conclusions. The authors make all substantive intellectual contributions.

## MOTIVATION OF PRECOGUI

Figure 6 reveals two critical patterns. First, the success rate (SR) declines sharply with an increasing number of injected disturbances. Reactive baselines plummet from nearly 100% SR to below 20% with zero to six injections, showing a performance gap of at least 10% by just two injections (left panel). This highlights the inherent brittleness of purely reactive policies under sustained interference. Second, disturbance timing significantly impacts performance (right panel). Shifting a single injection later in the trajectory yields greater SR losses across all baselines. For instance, UI-TARS exhibits an SR drop escalating from 3.0% (steps 0–5) to 16.4% ($> 20$ steps). In contrast, PRECOGUI demonstrates consistent resilience, increasing only from 1.6% to 7.1%—approximately $2.3\times$ less degradation than UI-TARS in the long-horizon tail—while maintaining higher nominal

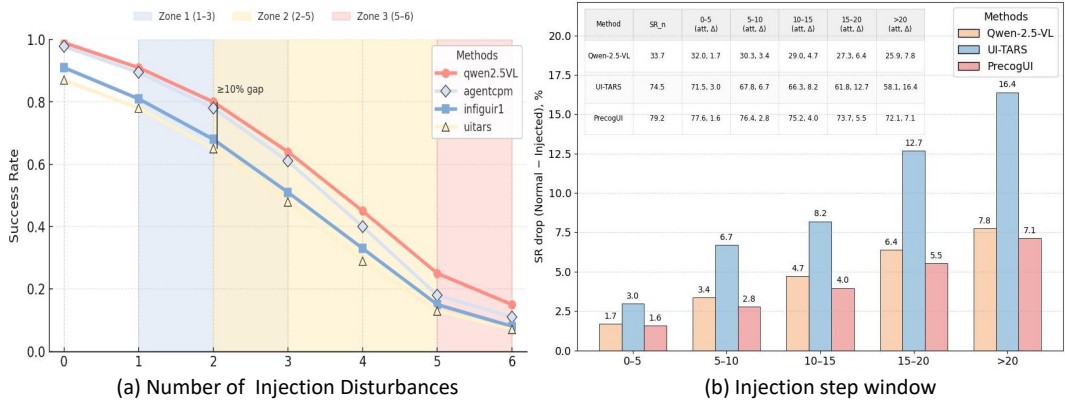

Figure 6: Impact of disturbance count and timing on policy success rates. The left panel shows SR degradation with increasing disturbance count. The right panel illustrates the greater sensitivity of reactive policies to later disturbance injections, in contrast to PRECOGUI's robustness.

SR. These trends suggest that coupling experience priors with look-ahead simulation is crucial for mitigating late-stage error cascades.

ANALYSIS OF LONG-HORIZON EVALUATION

## RELATED WORK

**Multimodal Large Language Models.** MLLMs (Li et al., 2023; Liu et al., 2024; Yue et al., 2024) have emerged as a central enabler for GUI automation, boosting both perceptual and reasoning capabilities of agents. By parsing complex screen structures and grounding natural language instructions in UI elements, MLLMs serve as the perception backbone for many mainstream agents (Wang et al., 2024a; Yang et al., 2025). However, existing MLLMs (Wang et al., 2024b; Chen et al., 2024; Lai et al., 2024) are primarily pre-trained or fine-tuned on static, single-turn perception tasks such as visual question answering (Ma et al., 2024) or image captioning (Dai et al., 2023b). Consequently, in dynamic UI scenarios, they tend to be stateless and myopic, producing immediate responses without sequential modeling or anticipatory reasoning.

**GUI agents.** Research on GUI agents (Gou et al., 2025b; Liu et al., 2025a; Xu et al., 2025a) has also explored diverse strategies for policy learning and grounding. A common paradigm (Lu et al., 2025; Lin et al., 2025) is to fine-tune multimodal models, mapping instruction and screenshot inputs into sequential action predictions. For example, UGround (Gou et al., 2025a) trains a purely visual grounding model on millions of UI elements, enabling click and operation solely through visual localisation. Recent efforts (Gao et al., 2025; Zhang et al., 2025a) have added structure and memory, with AutoDroid (Wen et al., 2024a) handling anomalies by learning corrective scripts and MapAgent retrieving layout traces during planning. While effective on short, static benchmarks (Gao et al., 2024), these methods (Lei et al., 2025; Xu et al., 2025b) remain confined to a reactive framework, in which agents make decisions based solely on the current observation, leaving them vulnerable to unforeseen perturbations. An unexpected pop-up can easily hijack the agents attention, while even minor loading delays may be misinterpreted as failed actions.

## POLYTOUCH: A MULTI-GESTURE AND MACRO EXECUTION LAYER

Real-world mobile applications often require multi-pointer and multi-step interactions, such as three- or four-finger system shortcuts, pinch/zoom and rotation in media and map viewers, or co-ordinated sequences in creative tools. Existing GUI agents generally assume single-touch atomic operations and one-shot execution, which makes them fragile when facing complex gestures, long interaction flows, or OS-level controls that demand precise synchronization. To address this gap, we introduce **PolyTouch**, an execution layer that extends the action space to multi-finger gestures and macro-level commands with explicit timing, guards, and rollback mechanisms.

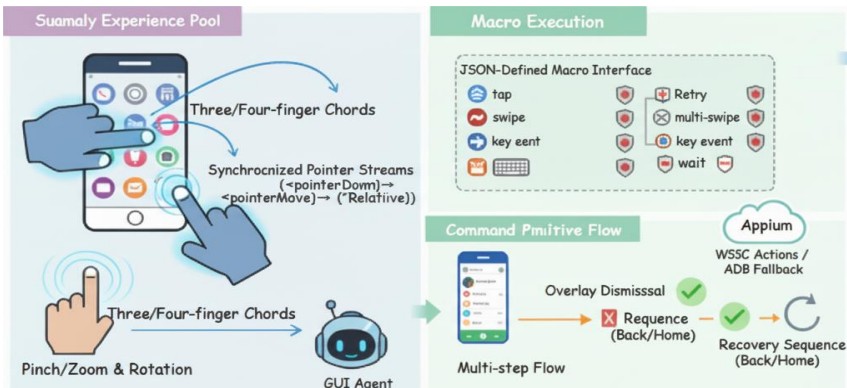

Figure 7: The illustration of PolyTouch, a multi-gesture and macro execution layer for GUI agents. It depicts multi-finger gestures and macro-level commands, highlighting their role in robust, long-horizon task execution.

PolyTouch supports a wide range of interaction patterns rarely considered in prior work: (i) Multi-finger chords for dialogs, split-screen, or editing shortcuts; (ii) Continuous gestures such as pinch, zoom, and rotation; (iii) Multi-step flows with explicit waiting, retries, and overlay dismissal; (iv) Recovery sequences (e.g., back, home, or targeted close) that must be executed atomically to exit unexpected states. These abstractions allow agents to operate robustly in long-horizon tasks where traditional atomic actions fail.

PolyTouch builds on Appiums W3C Actions API for deterministic multi-pointer synthesis and it falls back to ADB when accessibility channels are blocked. Its design centers on: (1) deterministic timing through tick-based scheduling; (2) unified coordinate formats (index, relative-in-box, absolute) with boundary-safe mapping; (3) a declarative macro interface that bundles taps, swipes, multi-swipes, key events, and waits into atomic, retryable units; (4) graceful degradation to equivalent ADB commands while preserving ordering and timing.

PolyTouch exposes two main capabilities: **(a) Multi-gesture execution.** Three- and four-finger gestures are represented as synchronized pointer streams (`pointerDown` → `pointerMove` → `pointerUp`), while pinch/zoom and rotation are parameterized around target boxes and derived from relative coordinates. **(b) Macro execution.** JSON-defined macros encapsulate an ordered list of primitives with explicit guard, retry, and rollback semantics, supporting flexible coordinate specifications.

PolyTouch integrates into the agent control loop by providing reliability-aware plans and structured execution reports (success flags, layout changes, anomaly tags). These outputs feed the Proactive Experience Pool to accumulate reusable patterns and guide the Pre-cognitive Execution Controller in anticipating failures and triggering recovery. In this way, PolyTouch transforms low-level taps into a closed-loop, macro-level control primitive that is both expressive and robust.

## ADDITIONAL EXPERIMENTS

### IMPLEMENTATION DETAILS

**Hardware & Devices.**    All experiments were conducted on a single training node with **8× NVIDIA H20 (96 GB)** GPUs. For on-device evaluation, we used a pool of mainstream Android phones covering **Huawei/Honor**, **Xiaomi/Redmi**, and **OPPO/realme**, spanning Android 10–14 and common resolutions (720p–1440p). Devices were connected over USB with ADB (USB debugging enabled) for reliable screenshot capture and input dispatch; Wi-Fi ADB was used only for long-duration soak tests.

**Data Collection & Real-World Tests.**    We employ **Appium 2.x** (Android driver: uiautomator2) together with ADB to (i) scrape view hierarchies and screenshots, (ii) execute action sequences in real apps, and (iii) log pre/post frames, timing, and outcomes for replayable

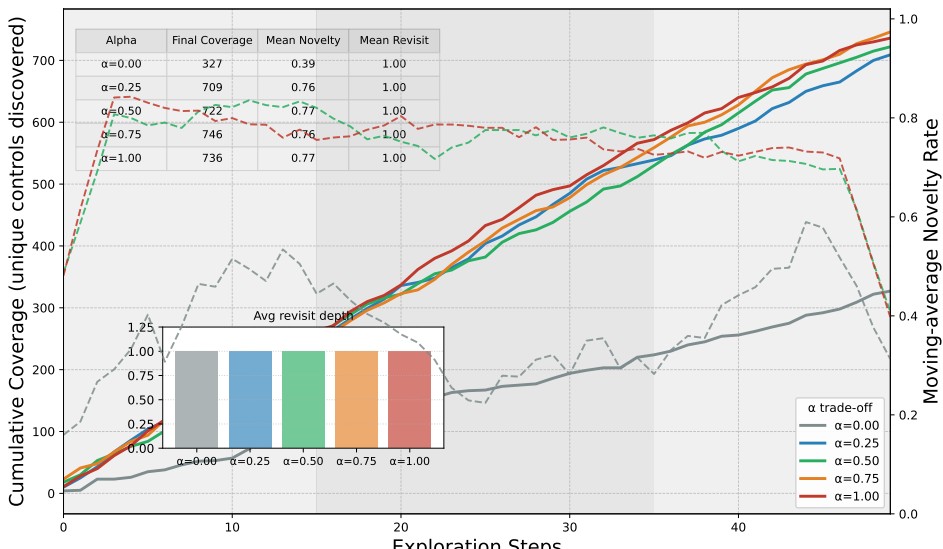

Figure 8: Effect of $\alpha$ in the exploration value over a 50-step horizon. The main curves show cumulative coverage of unique controls for different $\alpha$; dashed traces (right axis) depict the moving-average novelty, and the inset summarizes average revisit depth and final/mean statistics. Larger $\alpha$ prioritizes discovery and accelerates coverage, while smaller $\alpha$ favors rare-state revisits at the cost of slower expansion.

trajectories. For latency-critical fallback (e.g., when Appium is blocked by transient overlays), we issue low-level commands via `adb shell input` (tap/swipe/keyevent) and re-sync with Appium on the next stable frame. All experiments use fixed random seeds and identical capture settings across devices; screen coordinates are normalized to $[0, 1]$ and mapped to device pixels at runtime.

## BENCHMARKS

**Grounding-Centric Benchmarks: ScreenSpot Series.** Accurate element localization is the foundation of GUI automation. ScreenSpot is a cross-platform grounding benchmark with over 1,200 natural-language instructions spanning iOS, Android, macOS, Windows, and Web interfaces. Each instruction is paired with pixel-level bounding boxes and element-type labels (text, icon, or widget) and covers challenging scenarios such as icon-text composites and occluded controls.

**Navigation-Centric Benchmarks: AndroidControl & GUI Odyssey.** Once elements can be reliably located, agents must navigate within and across apps. AndroidControl(Li et al., 2024), the largest public mobile navigation corpus, contains 15,283 human demonstrations divided into low-difficulty single-app workflows (< 10 steps) and high-difficulty cross-app tasks with real-time interruptions (e.g., Select photo from Gallery Upload via Email). It evaluates agents comprehension of both high-level goals (Book a ride) and low-level operations (Tap Search). GUI Odyssey (Lu et al., 2024) extends this to long-horizon, cross-app navigation with 7,735 mission-based episodes across 201 apps and 1,400+ app combinations. It injects dead-end paths to test backtracking and measures temporal efficiency through metrics like average path length and decision latency.

**Disturbance-Aware Benchmark: InterfereBench.** InterfereBench covers 34 applicationscomplex games, enterprise tools, and general appswith bilingual (Zh/En) UIs recorded on diverse phone models. It contains 1,160 long-horizon trajectories (1437 steps) and 27,124 screenshots; we captured 574 real abnormal screens and curated 217 synthetic disturbances (pop-ups, notifications, black screens, layout shifts). Each task is paired with a clean baseline and perturbed variant(s) to enable controlled normal vs perturbed comparisons. Annotations include high-level goals and step-level structures (action type, normalized coordinates, UI boxes, screen deltas, outcomes).

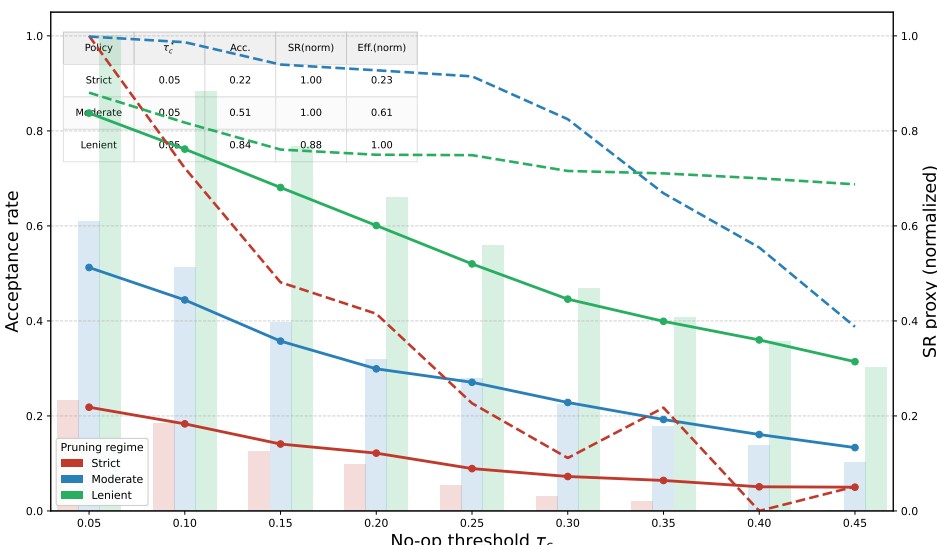

| Policy | $\tau_c$ | Acc. | SR(norm) | Eff.(norm) |
|---|---|---|---|---|
| Strict | 0.05 | 0.22 | 1.00 | 0.23 |
| Moderate | 0.05 | 0.51 | 1.00 | 0.61 |
| Lenient | 0.05 | 0.84 | 0.88 | 1.00 |

Figure 9: Parser-threshold analysis. The figure summarizes Stage-1 pruning versus $\tau_c$: acceptance monotonically decreases as $\tau_c$ grows; the *Moderate* regime offers the best acceptance-success-efficiency trade-off near $\tau_c \in [0.25, 0.35]$.

HYPERPARAMETER ANALYSIS OF THE EXPLORATION VALUE.

We study the single coefficient $\alpha$ that balances novel-control discovery against rare-state probing during exploration. As illustrated in Figure 9 On a 50-step horizon, larger settings (e.g., $\alpha \geq 0.5$) consistently deliver higher cumulative coverage and higher moving-average novelty, indicating faster expansion of the actionable UI space. Very small $\alpha$ emphasizes repeatedly visiting under-explored screens; while this can stabilize early behavior, it sacrifices coverage and slows progress. We observe no significant increase in redundancy within 50 steps, suggesting that short-horizon exploration benefits most from prioritizing discovery. In practice, $\alpha \in [0.5, 0.75]$ is a strong operating region that front-loads novel controls without noticeable revisit overhead. For longer horizons or highly volatile apps, an adaptive schedule is preferable: start near $\alpha \approx 0.5$ to stabilize initial navigation, then increase toward $0.75$–$1.0$ as the uncovered-control ratio declines. Overall, $\alpha$ provides an interpretable knob for exploration granularity; tuning (or scheduling) it materially impacts coverage speed and downstream success rates.

HYPERPARAMETER ANALYSIS OF PARSER THRESHOLDS.

We examine how the Stage-1 pruning thresholds (self-loop ratio and no-op ratio) interact with the no-op cutoff and impact downstream quality, as shown in Figure. 9. As the cutoff increases, the acceptance rate drops monotonically across all regimes (e.g., from ~0.60–0.65 at a low cutoff of 0.05 to ~0.20 at 0.45), indicating that more micro-changes are filtered as no-ops. *Strict* pruning rapidly depresses acceptance (often $< 0.25$ once the cutoff exceeds $\approx 0.20$), and downstream quality declines as data volume becomes the bottleneck. *Lenient* pruning maintains high acceptance ($> 0.55$ across most cutoffs) but retains many low-signal segments; the success proxy plateaus or degrades when the cutoff is high (e.g., normalized success $\lesssim 0.55$ once the cutoff $\geq 0.35$). By contrast, the *Moderate* regime achieves the best balance in a mid-range cutoff of **0.25–0.35**: acceptance stays around **0.35–0.50** while the normalized success proxy peaks around **0.75–0.85**, yielding the highest harmonic mean of acceptance and success.

**Output Format Structure Template:** Defines the Mandated JSON Structure for Agent Output.

{" **Historical_status**": "Success|Failed|Unknown - Evaluate if the previous action visually achieved its intended goal. Base this ONLY on the screen image. Ignore the execution result status provided in the input.",
  "**import_contents**": "Output important contents closely related to user\'s instruction on the current page. If there is, please output the contents. If not, please output empty string ".",
  "**think**": "Provide a step-by-step thinking process. Analyze the current screen, relate it to the overall task and the visual outcome of the previous step ('Historical_status'). Decide the next best *single* action. Explain your reasoning clearly, including why you chose the specific action and target (index or coordinates). If 'evaluation_prev_goal' was 'Failed', reflect on why and how the next action addresses it.",
  "**progress**": "Assess the overall progress. List the sub-tasks from the main `task` or `task_steps` that have been verifiably completed up to the current step. This judgment must be based on the *actual visual evidence* on the screen, not merely on the sequence of actions planned or executed, as actions can fail or produce unexpected results.",
  "**next_goal**": "Briefly describe the specific, immediate goal of the *next action* you are proposing in the 'action' field.",
  "action": {
    "**action_name**": { /* dictionary of parameters for the action */ } } }

{" **Historical_status** ": "Failed - The previous action (open settings) was blocked by a pop-up overlay.",
  "**import_contents**": "Popup dialog is visible with title '活动提醒'; two buttons: '立即前往' (index 15) and '稍后再说' (index 9). A close icon 'X' at top-right (index 5). Underlying Honor of Kings lobby is dimmed and not interactable.",
  "**think**": "A blocking popup covers the lobby. The overall task is to continue interacting with the lobby. The safest way to resume is to dismiss the popup without navigating elsewhere. The close icon at index 5 is visible and should remove the overlay without side effects. I will use the 'click' action with this index.",
  "**progress**": ["1. Reached the main lobby (visible in background)."],
  "**next_goal**": "Dismiss the popup to restore interaction with the lobby.",
  "**action**": {"click": {"position": 5}}}

Figure 10: Mandated JSON Schema for Agent Reasoning. The figure shows the output template (left) and an in-context example of handling a pop-up overlay (right).

## PROMPTS IN AUTOMATED PIPELINE

### OUTPUT FORMAT STRUCTURE TEMPLATE

As illustrated in Figure 10, our `Deep Think & Decision` mechanism is governed by a mandated JSON schema that structures the agent's output. This schema enforces a rigorous, multi-stage reasoning process through several key fields: `Historical_status` for visual verification of the previous action's outcome, severing reliance on potentially noisy execution logs; `import_contents` for grounding the agent's awareness in the current UI context; `think` for articulating a step-by-step causal rationale; `progress` and `next_goal` for explicit task decomposition and forward planning; and finally `action`, which specifies the precise, parameterized command for environmental actuation (e.g., via index-based coordinates). Crucially, the schema's emphasis on populating fields like `Historical_status` based *solely* on visual evidence establishes a tight closed-loop verification system. This structured output thereby functions as a transparent and auditable interface between the agent's cognitive deliberation and its concrete actions within the GUI environment.

### ACTION SELECTION TEMPLATE

To ensure robust action grounding, we define a hierarchical, three-tiered schema for specifying target coordinates, enforcing a graceful degradation from semantic to pixel-level references. The primary and most preferred format is **(1) Highlight Index**, which targets an element via a unique semantic identifier, providing high resilience to minor layout shifts. The secondary format, **(2) Relative-in-Box**, is used for sub-point targeting within an indexed element, thus combining a semantic anchor with fine-grained precision. The final fallback, **(3) Absolute Coordinates**, is used only when semantic indexing is infeasible, targeting a point in a normalized coordinate space. This strict priority order '(1) > (2) > (3)' ensures that the agent always defaults to the most robust targeting method available.

### ANOMALY HANDLING TEMPLATE

As shown in Figure 12, we frame anomaly handling as a concise, cross-task routine over prediction and verification. Given the current layout $L_t$ and the forecast $\hat{L}_{t+1}$, the agent applies fast rules to classify and mitigate: (i) Pop-up/Overlaydismiss via safe affordances

**Action Position Selection:** Standardize the three mutually exclusive ways to target UI and set a priority order.

When specifying the target for an action, choose EXACTLY ONE 'position' form:

1) **Highlight Index:** "position": <int>
   - Preferred when the target reliably maps to a single highlighted box.

2) **Relative-in-Box:** "position": [<int>, <float>, <float>]
   - Use when the target is INSIDE the highlighted box but needs a precise sub-point (e.g., a small icon inside a large button).
   - Floats are relative coordinates within that box in [0.0, 1.0]:
     (0,0)=top-left, (1,1)=bottom-right.

3) **Absolute Center Coordinates:** "position": [<int>, <int>]
   - Fallback when no reliable highlight exists, the index is unreadable, or the box is inaccurate/too large.
   - Coordinates are normalized pixels in [0,1000] for (x,y); values must not exceed bounds.

Priority: (1) > (2) > (3). If you use (2) or (3), briefly justify why in your reasoning.

"click": {"position": 12}

"click": {"position": [7, 0.85, 0.25] }

"click": {"position": [642, 318] }

Figure 11: Action command selection adopts a three-tiered, prioritized format (Index → Relative-in-Box → Absolute), with in-context examples: (1) semantic targeting via a unique index; (2) fine-grained targeting using relative coordinates within an indexed element; and (3) a robust fallback to normalized absolute coordinates.

**Anomaly Handling:** Cross-task discipline for reasoning, decomposition, verification, and termination.

**A) Inputs:**
- $L_t$: current symbolic layout inferred from the current screen.
- $L_{t+1}$: predicted next-step layout given (L_t, candidate action, goal).
**B) Fast anomaly rules** (apply to either the current screen or \hat{L}_{t+1} when available):
*1) Pop-up/Overlay Anomaly:*
- Signs: High-z modal panel covering main content; overlaps multiple interactive controls; typical dismiss affordances ("X/Close/Cancel/Not now/Later"), dimmed background.
- Mitigation: Click a safe dismiss (X/Close/Cancel/Later) → if none, try back once → short wait and re-check. Avoid "Go now/Claim/Start trial" unless explicitly required. Blank-Screen Anomaly:
*2) Blank Screen*
- Signs: Almost no interactive elements or very low saliency; prediction also "blank".
- Mitigation: Short wait → if persistent, back once or refresh per platform → optionally return to a known stable page (menu/home).
*3) Freeze / Ineffective Action*
- Signs: Layout nearly unchanged and intended state not updated; animation halts without transition.
- Mitigation: Retry once with improved targeting (Relative-in-Box or safer index) → if still unchanged, back or wait then retry → if recurrent, re-plan (alternate path/control).
*4) Off-Goal / Misdirection*
- Signs: Next screen diverges from goal (e.g., store/ads), goal elements vanish.
- Mitigation: Abort the risky path; dismiss/ back to restore context; choose a safer, on-goal alternative.
**C ) Post-Mitigation Re-check:** After handling any anomaly, **re-check**: target page/controls are visible and **no overlay remains**; then continue the main task.

Figure 12: Mandated JSON Schema for Agent Reasoning. The figure shows the output template (left) and an in-context example of handling a pop-up overlay (right).

(X/Close/Cancel/Later); (ii) Blank Screen short wait, then Back/refresh or return to a stable hub; (iii) Freeze/Ineffective Actionsingle retry with safer targeting (Relative-in-Box or safer index), else Back/re-plan; (iv) Off-Goal/Misdirectionabort the path and restore on-goal context. A compulsory post-mitigation re-check gates progress: continue only when target controls are visible and no overlay persists.

ROLE AND CONTEXT TEMPLATE

To structure the agent's operational context, we define a clear set of responsibilities and a standardized input format for each reasoning step. As illustrated in Figure 13, the agent is prompted with persona as an expert GUI automation agent. For each step, it receives a tripartite input: (1) the current screenshot augmented with indexed, highlighted bounding boxes over interactable elements; (2) feedback on the execution status (e.g., success or failure) of the prior action; and (3) the current temporal step index. Crucially, the agent is explicitly instructed to ground its reasoning *solely on visual evidence*, judging task progression based on observable changes in the UI state rather than

> **Role & Context:** Define the agent's responsibilities and the I/O context (screenshots with highlighted regions, prior execution result, step number).
>
> You are an expert GUI automation agent. Your job is to complete the user's task by interacting with PC/mobile GUIs using screenshots.
>
> For each step, you receive:
> 1) The current screenshot with highlighted UI regions (each region has a top-left index).
> 2) The previous action's execution result (success/failed/unknown).
> 3) The current step number.
>
> Always reason from on-screen visual evidence. Highlight boxes help locate elements; completion must be judged by actual page state changes (texts, titles, control states).

Figure 13: Role and context template. Specifies agent responsibilities and I/O context with indexed screenshots, prior execution results, and step numbers to guide evidence-based task completion.

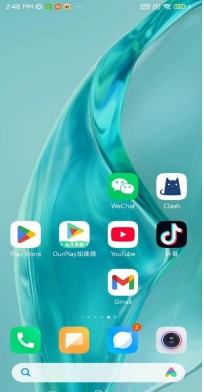

> **OS-Specific Hints:** Encode platform conventions (home screens, recent apps, ADB keyboard, games/back behavior).
>
> **Android Hints:**
> - To find apps, swipe left/right on home screens. When starting an app, click the CENTER of the app icon (not its label). If required by your action schema, set 'open_app': true.
> - Special keys: 'home', 'back', 'recent'. 'recent' opens the app switcher.
> - If the ADB keyboard is visible, the input field is active: do NOT provide 'position'; directly 'input_text'.
> - In games, 'back' may be ineffective; follow in-game flows. GM commands are valid only after entering the game and must follow the provided order strictly.
> - On the recent apps screen, to kill an app, swipe its card off-screen (end point beyond the screen edge).

Figure 14: OS-specific action hints. Encodes Android conventions for app access, navigation keys, keyboard input, in-game flows, and app termination to ensure robust, context-aware execution.

uncritically accepting the programmatic execution status. This mandate establishes a tight, closed-loop visual verification process for all decision-making.

## OS-SPECIFIC HINTS

As shown in Figure 14, we encode platform conventions into structured hints that guide robust action execution on Android. These rules address common UI operations and context-sensitive behaviors: (i) app launching via centered icon clicks with optional `open_app` flag; (ii) special system keys such as `home`, `back`, and `recent` for navigation control; (iii) text input handling by directly invoking `input_text` when the ADB keyboard is active, avoiding redundant position specifications; (iv) game-specific flows where the `back` key may be ineffective, requiring strict adherence to in-game command order; and (v) app termination through swipe-off gestures in the recent-apps screen. Collectively, these hints ground agent actions in OS-level semantics, reducing execution ambiguity and improving cross-context stability.

## GENERAL INSTRUCTIONS.

As shown in Figure 15, this template encodes cross-task discipline for structured reasoning and verifiable execution. It emphasizes (i) step-by-step task decomposition into checkable sub-steps; (ii) precise targeting using highlighted regions or indices while avoiding ambiguous clicks; (iii) progress verification strictly by on-screen evidence such as titles, messages, or control states; (iv) controlled waiting to accommodate delays or animations; (v) fallback to anomaly-handling rules when overlays appear; and (vi) termination only after explicit visual confirmation of success. When targeting remains uncertain, the agent is required to re-locate or choose safer alternatives, ensuring robustness

against cascading errors. Collectively, these rules establish a disciplined action loop where correctness validation precedes task advancement.

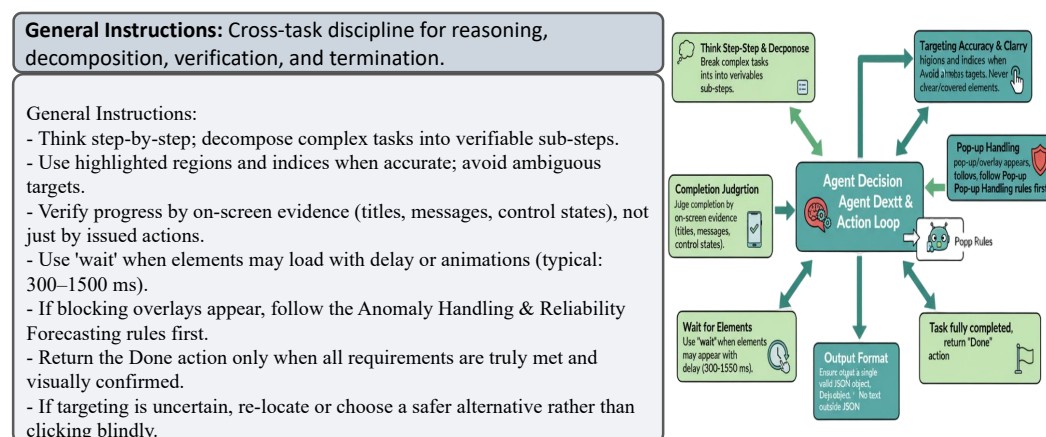

Figure 15: General instruction template. Defines structured reasoning, precise targeting, verification, controlled waiting, and disciplined termination to ensure robust, evidence-driven task execution.

## QUALITATIVE ANALYSIS

### APPS POP-UP HANDLING

As shown in Figure 16, we deploy a type-aware policy that closes in-app pop-ups while preserving task context. The controller first classifies the pop-up(i) announcement/notice panels, (ii) gift-package ads, (iii) event promotions, or (iv) confirmation/input dialogsand selects the safest dismiss affordance. Execution follows our hierarchical position schema: prioritize element *indices* for X/Close/Cancel/Later; degrade to *Relative-in-Box* when the target is a sub-control; and use *normalized absolute* coordinates only when indexing is unreliable. Each thumbnail shows the predicted command (index or relative point) rendered beneath the image; progress continues only after the overlay is visually cleared.

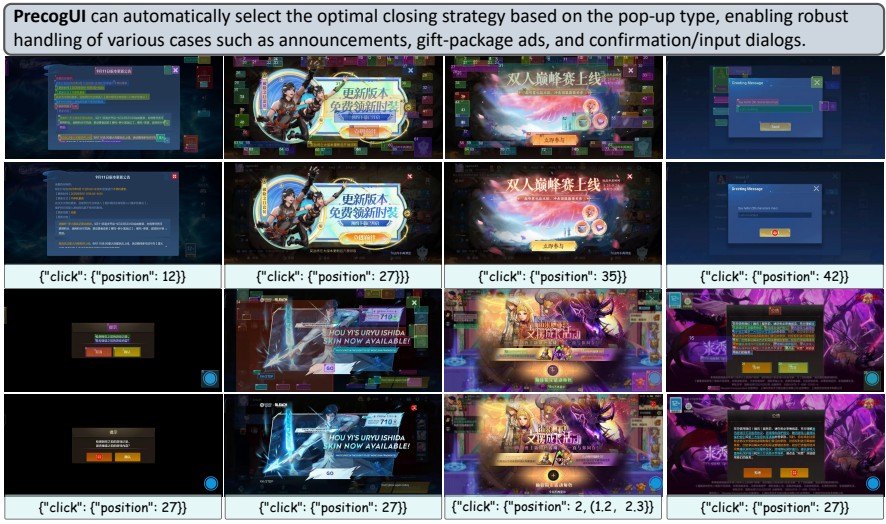

Figure 16: Apps pop-up handling. A type-aware policy combined with hierarchical position selection (Index → Relative-in-Box → Absolute); the figure presents concrete dismissal commands for diverse pop-up cases.

SYSTEM-LEVEL POP-UP HANDLING.

As shown in Figure 17, we handle OS-mediated interruptionssystem notifications, risk alerts, and permission requestsvia a type-conditioned, safety-first policy. The controller classifies the pop-up and selects the safest affordance (e.g., `Cancel/Close`, `Allow only while in use`, `Deny`). Execution uses our hierarchical position scheme, prioritizing element *indices* and backing off to *Relative-in-Box* or normalized *Absolute* coordinates only when indexing is unreliable. Each panel displays the issued command (primarily index clicks), and progress resumes only after the overlay is visually cleared to preserve task context.

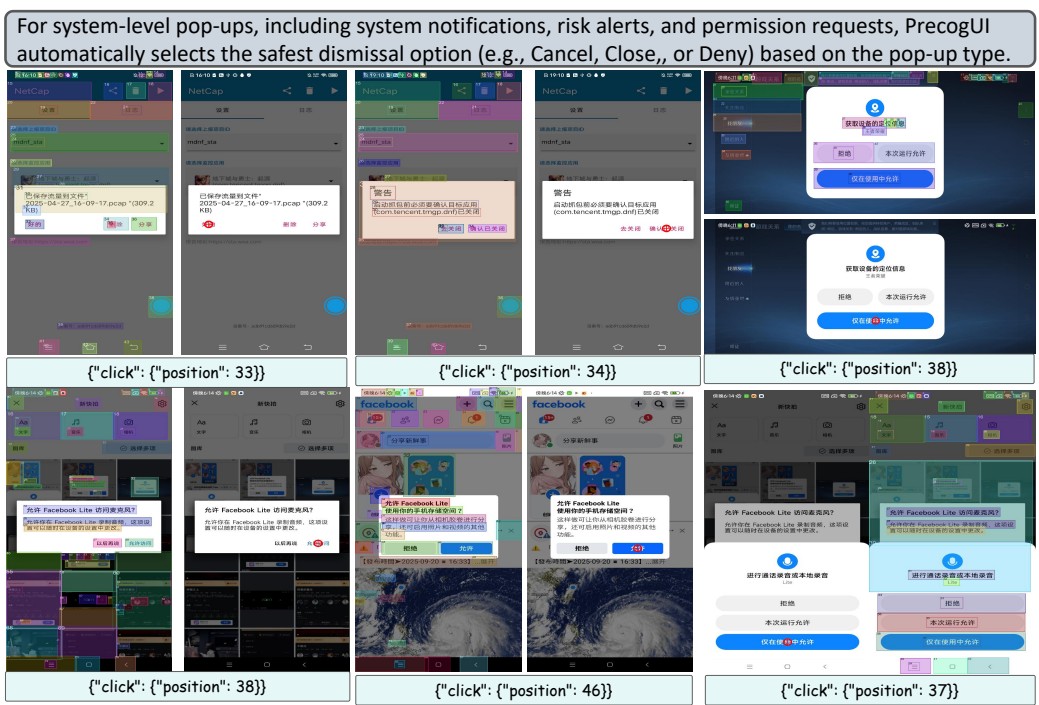

Figure 17: System pop-up handling. A type-aware policy selects safe dismissal actions and executes them with index-prioritized targeting; the figure shows concrete commands for notifications, risk alerts, and permission requests.

ENVIRONMENT PERTURBATION HANDLING.

As shown in Figure 18, we address environment-level disturbances(black/white screens, loading delays, and network stalls) with a lightweight stabilization routine. Detection relies on low-saliency/blank frames, near-identical consecutive layouts, or stalled progress indicators. Mitigation is minimal yet effective: inject a short wait (e.g., 200 ms) to absorb transient transitions, then issue a single index-prioritized safe retry of the previous action; progress resumes only after visual evidence of recovery, otherwise control is escalated to the general anomaly rules.

**Layout-Shift Perturbations.** As shown in Figure 19, we address orientation/gravityinduced reflows (portrait ↔ landscape) with an orientation-aware re-localization routine. Upon detecting a layout shift (aspect-ratio change and index invalidation), the agent reconstructs the symbolic layout hash, re-indexes targets, and remaps the current goal to the new arrangement by type/text cues. Execution then follows the hierarchical position policy (Index → Relative-in-Box → Absolute), and progress is gated by visual re-check to ensure the intended control is active after rotation.

ADDITIONAL DISCUSSIONS

Forecasting future layouts is central to PrecogUI: look-ahead turns reactive observeact behavior into risk-aware planning that preempts pop-ups, freezes, and off-goal drifts, improving long-horizon sta-

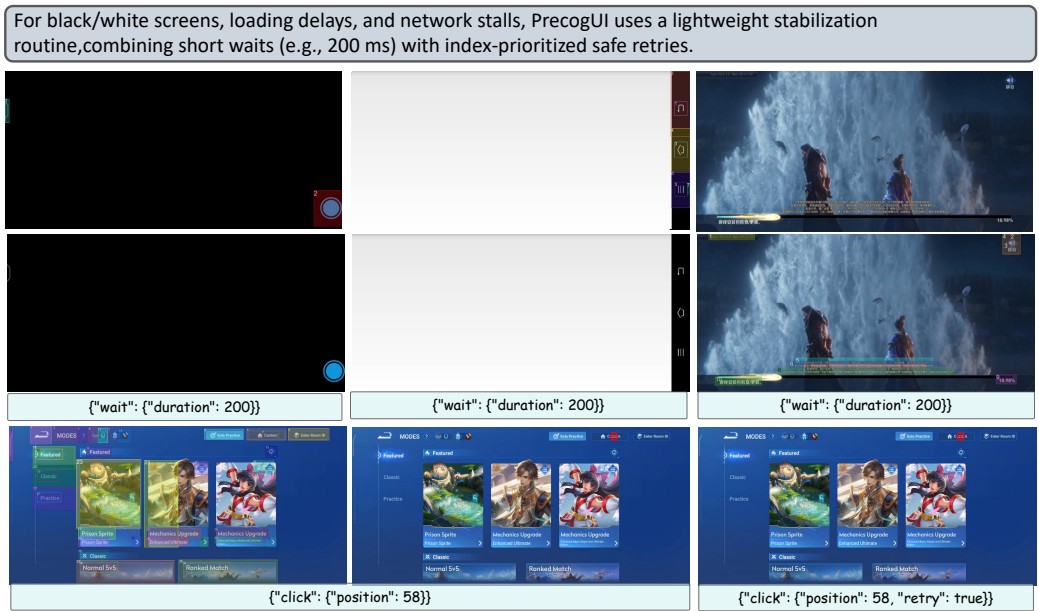

Figure 18: Environment disturbances. A lightweight routineshort waits plus index prioritized safe retries stabilizes black/white screens, delayed loads, and network stalls; the figure shows concrete `wait` and `retry` commands for representative cases.

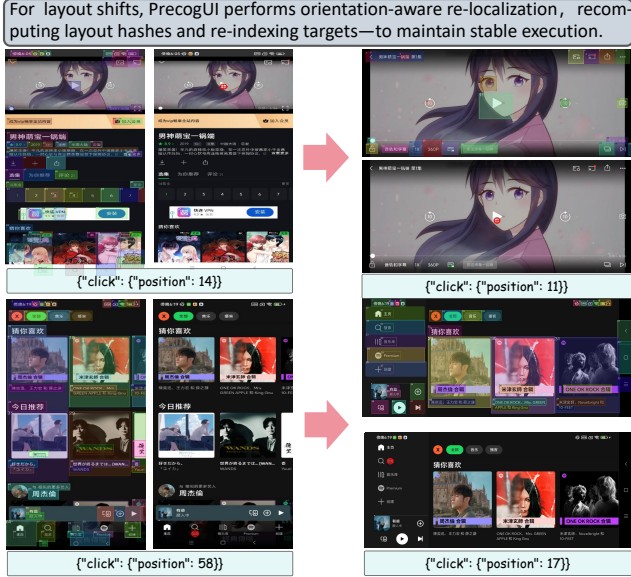

Figure 19: Layout-shift handling. PrecogUI rebuilds layout hashes and re-indexes targets under portrait/landscape transitions, executing with index-first targeting; the figure shows before/after screens with preserved action intent.

bility. However, timeliness is a key constraint. Pre-execution simulation and verification add latency and compute, which can be costly for real-time use or very long tasks. In addition, experience priors can become stale as apps update; outdated remedies hurt reliability unless memory is refreshed. Future work should adopt lightweight, anytime forecasting and drift-aware memory maintenance to preserve the gains of look-ahead without sacrificing responsiveness.

