# OpenReview forum: "PrecogUI: Proactive GUI Agents via Pre-cognitive Simulation and Experience Retrieval"
_ICLR.cc/2026/Conference — ICLR 2026 Conference Withdrawn Submission_

### Official Review · Reviewer_osCS · 2025-10-24

**Soundness:** 3
**Presentation:** 3
**Contribution:** 3
**Rating:** 4
**Confidence:** 4

**Summary:**

The paper introduces a proactive framework for GUI agents designed to move beyond traditional reactive behavior. Instead of responding only to the current interface state, the proposed system predicts possible future interface changes and recalls prior experiences to make more reliable decisions. This proactive mechanism allows the agent to anticipate disturbances such as pop-ups, delays, or layout changes, and to adjust its actions before errors occur. The authors also build a new benchmark with long-horizon and interference-rich scenarios to evaluate robustness. Experimental results demonstrate that the proposed approach significantly improves success rates and stability compared with existing reactive baselines, highlighting the effectiveness of predictive reasoning and experience-based control in complex GUI environments.

**Strengths:**

- The paper addresses an important and practical problem by enabling GUI agents to act proactively rather than reactively.
- The proposed framework effectively integrates predictive simulation and experience retrieval, forming a well-motivated and coherent design.
- A new benchmark for long-horizon and interference-rich tasks is introduced, providing a solid basis for evaluating robustness.
- Experimental results show clear and consistent performance gains over competitive baselines.
- The ablation and visualization analyses are thorough, clearly illustrating the contribution of each component.

**Weaknesses:**

- The proactive mechanism introduces additional computational overhead, and the paper does not clearly quantify its efficiency trade-offs in real-world settings.
- The ablation and visualization studies, though useful, remain mostly qualitative, more quantitative analysis on prediction accuracy or memory retrieval efficiency would strengthen the claims.
- While the proposed benchmark is comprehensive and well-structured, it is largely synthetic and relies on controlled interference injection. As a result, it may not fully capture the diversity and unpredictability of real-world GUI environments, such as variations across devices, localized interfaces, or spontaneous user-driven interactions.

**Questions:**

- The paper repeatedly uses the term “pre-cognitive” or “proactive reasoning,” but the conceptual distinction between this and traditional planning or predictive control is not clearly formalized. How exactly is “proactivity” quantified or evaluated in experiments?
- The description of the “conditional diffusion model” for predicting future UI layouts lacks sufficient details. e.g., what conditioning signals are used (visual embeddings, action tokens, or both)? How is temporal consistency between consecutive predictions maintained
- It’s not specified whether the baselines were retrained or fine-tuned under the same disturbance conditions. Were all models given equal access to environment feedback and screenshots?

---

### Official Review · Reviewer_XcCT · 2025-10-30

**Soundness:** 2
**Presentation:** 2
**Contribution:** 2
**Rating:** 4
**Confidence:** 4

**Summary:**

This paper introduces PrecogUI, a framework for proactive GUI agents that move beyond reactive policies by integrating offline experience retrieval, look-ahead simulation, and exception-aware execution. Its three core components—Proactive Experience Pool (PEP), Proactive Simulation Executor (PSE), and Pre-cognitive Execution Controller (PEC)—are designed to enhance agent robustness in long-horizon, disturbance-prone environments, mitigating cascading failures from anomalies such as pop-ups and layout perturbations. The authors build AutoTraj, a data generation engine, and release InterfereBench, a new benchmark targeting challenging GUI scenarios. Extensive experiments and ablations demonstrate that PrecogUI outperforms existing baselines in both robustness and grounding/generalization tasks.

**Strengths:**

1. The proposed system covers all essential facets of robust GUI agent operation: memory-based experience retrieval, proactive symbolic UI simulation, and a closed-loop controller for recovery and action selection. Figure 2 and Figure 3 visually clarify the interplay between PEP, PSE, and PEC, concretely illustrating how these modules coalesce into a cohesive whole.

2. The work features rigorous evaluation across both the introduced InterfereBench and public datasets (AndroidControl, GUI-Odyssey, ScreenSpot), including clear comparisons with strong baselines (GPT-4o, Qwen-2.5-VL, UI-TARS-1.5, etc.). Table 1 and Table 3 provide concrete evidence for PrecogUI’s superior robustness and success rates, particularly under strong perturbation.

**Weaknesses:**

1. The paper overlooks several highly relevant recent works directly related to benchmarks, proactive GUIs, process reward, self-supervised RL, and neuro-symbolic approaches in GUI grounding and control. Notably, none of the ScienceBoard, GUI-Shift, GUI-PRA, Mobile-Bench-v2, or NAVER/NEUSIS/HYDRA papers are cited or addressed. This undermines the positioning of PrecogUI as “first” or “comprehensive,” and weakens claims regarding novelty and benchmark comparison. This is particularly problematic given the existence of dynamic multitask GUI benchmarks, learning frameworks with explicit anticipation or reward shaping, and neuro-symbolic trajectories—all central to the scope here.

2. In Section 2.4, success probability is set directly equal to normalized layout dissimilarity (Eqn. for $\mathcal{D}_{\text{layout}}$). However, this mapping is questionable—major layout changes might not always imply positive action outcomes (e.g., transitions that lead away from the task goal or result in anomalies), and vice versa. There is no explicit calibration or empirical justification for using this measure as an action “success” likelihood, nor is sensitivity to this policy explored in depth.

3. How PrecogUI fares across application domains, UI complexity, or with specific disturbance types (other than aggregate overlays/environment disruptions shown in Figure 1(b)), especially in comparison to methods like GUI-Shift or Mobile-Bench-v2.

4. Class-level ablations (e.g., pop-up handling vs. layout-shift vs. freeze vs. blank screen) and whether the experience pool overfits to certain anomaly types.

5. Insufficient commentary on the memory update mechanism—how often must the PEP be refreshed in practice for evolving apps, and how does staleness impact recovery rates?

6. Insufficient Theoretical Justification and Critical Opaqueness in Some Modules: The “closed-loop” PEC controller is given via Algorithm 1 as a mixture of rule-based and model-driven logic. However, there is limited discussion of convergence guarantees, complexity of recovery loops (how often can actions oscillate or retry), or the implications of using an MLLM for verification. There is no empirical analysis of worst-case error accumulation or failure cases (other than a few qualitative samples).

7. Despite promising results in Table 3, it is not clear how well the approach generalizes beyond the test suite—e.g., unseen device types, radically different UI paradigms, or adversarially crafted anomalies. The benchmark design (InterfereBench) is strong but could be further discussed in terms of coverage.

8. While Figure 6 (Appendix) and qualitative figures showcase robustness improvements, the main text lacks a quantitative assessment of run-time trade-offs. Look-ahead simulation, memory querying, and model-based rollout introduce overhead—how scalable or real-time is the final system, and is there a performance penalty for very long-horizon tasks? This is only briefly acknowledged in passing as a limitation.

9. Details and Hyperparameter Settings Largely Deferred to Appendix: Critical implementation and hyperparameter settings (e.g., for AutoTraj, parser thresholds, diffusion noise schedule, and PEP update dynamics) are detailed in the supplement, potentially hampering reproducibility from the main text alone.

**Questions:**

see above

---

### Official Review · Reviewer_7LgD · 2025-11-03

**Soundness:** 2
**Presentation:** 2
**Contribution:** 2
**Rating:** 2
**Confidence:** 4

**Summary:**

The authors propose PrecogUI, a framework designed to enhance the robustness of GUI agents in long-horizon, disturbance-prone settings. The architecture is composed of three main components: a Proactive Experience Pool (PEP) for retrieving known anomaly and success interaction patterns , a Proactive Simulation Executor (PSE) to forecast UI state transitions and associated risks, and a Pre-cognitive Execution Controller (PEC) to integrate these modules and manage the decision-making loop. The paper also introduces InterfereBench, a new benchmark for evaluating agent robustness against sustained perturbations, which is generated by an automated pipeline AutoTraj.

**Strengths:**

- The InterfereBench benchmark, along with the AutoTraj data generation engine, is a contribution to the community for evaluating agent robustness in dynamic long-horizon environments.

**Weaknesses:**

- In L126, the authors claim that Proactive Experience Pool (PEP) forms a graph-structured memory of anomaly/success patterns via layout hashing and nearest-neighbor retrieval. However, in Section 2.3, no graph-structure is specified in the method description. In addition, the Retrieval Mechanism in PEP is hash-based, which is contradictory to Nearest-Neighbor Retrieval.
- In L251, the authors state that "Our premise is that success correlates with salient layout shifts." This assumption lacks theoretical justification or empirical validation. Thus, the subsequent claim "We then directly use this dissimilarity score as the success probability" is ungrounded.
- The reproducibility of the proposed framework is limited due to missing methodological details. For example, in the PEP module, the hash function $h_s$ is not defined. In the Proactive Simulation Executor (PSE) module, the specific conditional latent diffusion model used in the experiment, along with its denoising steps and pre-specified noise-schedule parameters, are not reported.
- The experimental evaluation lacks a critical baseline: a reactive agent using the Gemini-2.5-Pro backend, fine-tuned on InterfereBench.

**Questions:**

- What is the layout-difference measure $\mathcal{D}_{cal}$ shown in Eqn.2 used in the data construction stage? Is this metric identical to the one defined in Eqn.4?
- Could the authors empirically validate the robustness of the hash-based Proactive Experience Pool module?
- Could the authors clarify the exact specification of the (type, bbox) tuple used for layout hashing (L219)?
- In L246, the author states that "we apply a set of efficient rules to the predicted layout for anomaly recognition." Could the authors elaborate on these rules in detail?
- In Algorithm 1, what specific models do $\pi_{LLM}$ and MLLM correspond to in the experiments?
- In Fig. 9, the legends are somewhat unclear. What do the bar plot, solid lines, and dashed lines represent, respectively?
- Could the authors provide anomaly prediction metrics (e.g., precision/recall) to quantitatively validate the forecasting capability of the PSE?
- Could the authors report the total GPU hours to train the conditional diffusion model in the PSE module?
- What is the computational overhead and latency of the PrecogUI compared to other baselines?

---

### Official Review · Reviewer_zbha · 2025-11-09

**Soundness:** 2
**Presentation:** 2
**Contribution:** 2
**Rating:** 4
**Confidence:** 3

**Summary:**

This paper introduces PrecogUI, a proactive GUI agent architecture designed to improve robustness in long-horizon, dynamic interaction tasks, particularly where disruptive anomalies and non-goal cues (e.g., pop-ups, notifications) pose challenges for reactive agents. PrecogUI integrates three main components: a Proactive Experience Pool (PEP) for caching success/anomaly patterns, a Proactive Simulation Executor (PSE) leveraging conditional diffusion modeling to forecast symbolic UI layouts under candidate actions, and a Pre-cognitive Execution Controller (PEC) that fuses priors, simulation outcomes, and online feedback for closed-loop planning, anomaly handling, and execution recovery. The authors also propose InterfereBench, a new benchmark featuring long-horizon, perturbation-rich GUI trajectories, and demonstrate significant improvements in robustness and reliability over existing methods on both InterfereBench and public datasets.

**Strengths:**

1. Problem framing & scope. Robustness to disturbances in long-horizon GUI tasks is timely and clearly motivated; the paper formalizes state–action–result tracking and proposes a layout dissimilarity metric used by the controller.

2. Benchmarking contribution. InterfereBench and AutoTraj appear substantial—covering normal vs. perturbed conditions, and describing how realistic failures are injected.

3. Empirical breadth. The paper reports strong, consistent gains on InterfereBench and established benchmarks (AndroidControl, GUI-Odyssey, ScreenSpot), with tables that separate typical vs. adversarial settings.

**Weaknesses:**

1. Positioning vs. proactive agents. Related work on proactive/deliberative GUI agents is placed mostly in the appendix, which makes it hard to assess novelty relative to recent systems that combine experience, simulation, or multi-step planning.

2. Heuristic reliance in PEC. The controller fuses learned policies, LLM judgments, and several thresholds (e.g., candidate pruning, risk cutoffs). The paper gives the loop but not a principled procedure for choosing or adapting these thresholds across apps/tasks, raising concerns about brittleness and portability.

3. Scalability/latency left unquantified. The Discussion acknowledges overhead and staleness risk for the offline experience pool, but runtime/compute measurements are absent; this matters for practical mobile deployment.

**Questions:**

1. Runtime & resources: Please report step-time breakdown (PEP retrieval, PSE sampling, PEC loop), memory footprint, and mobile vs. desktop runtimes under typical horizons.

2. Out-of-distribution (OOD) stress tests: Can you evaluate on unseen apps/disturbances that fall outside InterfereBench’s taxonomy to show open-world robustness?

---

### Official Review · Reviewer_caRX · 2025-11-10

**Soundness:** 2
**Presentation:** 2
**Contribution:** 3
**Rating:** 6
**Confidence:** 2

**Summary:**

PrecogUI is  GUI-agent framework that combines (i) a system that caches success/anomaly state-action patterns, (ii) an executor that predicts the next symbolic UI layout to estimate success/anomaly risk, and (iii) a controller that fuses priors and predictions to act and recover. Further, the authors also introduce AutoTraj and InterfereBench to study robustness under injected perturbations (overlays, black screens, freezes). On InterfereBench, PrecogUI shows marginal drops in performance signifying its strength.

**Strengths:**

- Robustness benchmark: InterfereBench and AutoTraj provides paired clean/perturbed trajectories with realistic disturbances. It is a useful testbed for long-horizon disturbance robustness.

- Using a look-ahead layout predictor to pre-score candidate actions and detect anomalies before they happen is well-motivated for dynamic mobile UIs.

**Weaknesses:**

- Test-time fairness in experiments: The paper does not report tokens per step, steps per episode for PrecogUI vs. baselines, especially on InterfereBench where perturbations lengthen trajectories. Without compute-normalized curves, it is hard to tell whether robustness gains come partly from higher inference/decision budgets.

- AutoTraj’s Stage-2 uses an MLLM to rank trajectories. A human evaluation is needed to ensure there is no variance introduced due to LLM-as-a-judge setup.

**Questions:**

Please see weakness section.

---

### Note · Authors · 2025-11-14

I have read and agree with the venue's withdrawal policy on behalf of myself and my co-authors.